# Semi-Supervised Cell Instance Segmentation for Multi-Modality Microscope Images

**Ziyue Wang**[*]
School of Computer Science and Technology
Harbin Insititude of Technology (Shenzhen)
Shenzhen, China 518055
`200111326@stu.hit.edu.cn`

**Zijie Fang**[*]
Tsinghua Shenzhen International Graduate School
Tsinghua University
Shenzhen, China 518055
`fzj22@mails.tsinghua.edu.cn`

**Yang Chen**
Tsinghua Shenzhen International Graduate School
Tsinghua University
Shenzhen, China 518055
`cy21@mails.tsinghua.edu.cn`

**Zexi Yang**
School of Computer Science and Technology
Harbin Insititude of Technology (Shenzhen)
Shenzhen, China 518055
`200111318@stu.hit.edu.cn`

**Xinhao Liu**
School of Computer Science and Technology
Harbin Insititude of Technology (Shenzhen)
Shenzhen, China 518055
`200111304@stu.hit.edu.cn`

**Yongbing Zhang**[†]
School of Computer Science and Technology
Harbin Insititude of Technology (Shenzhen)
Shenzhen, China 518055
`ybzhang08@hit.edu.cn`

## Abstract

Many clinical and biological tasks depend on accurate cell instance segmentation. Currently, the rapid development of deep learning realizes the automation of cell segmentation, which significantly decreases the workload of clinicians and researchers. However, most existing cell segmentation frameworks are fully supervised and modality-specific. Towards this end, this paper proposes a semi-supervised cell instance segmentation framework for multi-modality microscope images. Firstly, $K$-Means clustering is utilized to discriminate the image modality. Then, for phase contrast and differential interference contrast images, Cellpose is adopted. For brightfield images, we subdivide them into two sub-categories according to the cell diameter by $K$-Means and optimize a U-Net for the large

---

[*]Ziyue Wang and Zijie Fang contributed equally.
[†]Corresponding author

36th Conference on Neural Information Processing Systems (NeurIPS 2022).

diameter group. For fluorescence images, we propose a semi-supervised learning strategy using CDNet. The leaderboard shows that our proposed framework reaches an F1 score of 0.8428 on the tuning set, which ranks 6th among all teams.

# 1 Introduction

Cell segmentation is the prerequisite for many diagnostic and biological tasks [1, 2]. Accurate cell segmentation not only improves the clinicians' and researchers' qualitative understanding of the change in cell distribution in different diseases but also enhances their comprehension of how different diseases affect the cell number, diameter, direction, and other texture features quantitatively. Therefore, cell segmentation advances researchers' and clinicians' understanding of disease pathogenesis, which helps them achieve more accurate diagnosis and treatment options [3, 4, 5].

Traditionally, pathologists have to identify and analyze cells manually under a microscope. However, due to the dense cell distribution and the heterogeneity of cell morphology, it is very time-consuming and laborious to observe cells by eyes [6]. The digitalization of pathological images has promoted the emergence and development of computational pathology, making it possible to use computer vision methods such as deep learning to realize automated cell segmentation, which significantly facilitates the pathologists and researchers and reduces their workload [7].

However, most of the existing automated cell segmentation methods are based on fully supervised frameworks, which require accurate cell segmentation masks for training [8, 9]. Due to the enormous scale difference between the pathological tissue slides and the cells, it is a complicated and laborious task to perform accurate cell annotation [10, 11]. In addition, due to the professionality of pathological images, it is not feasible to utilize distributed annotation strategies popular in natural image annotation, such as crowdsourcing [12, 13]. In order to solve this problem, some researchers tried to train cell segmentation networks under a semi-supervised paradigm, which uses a small amount of labeled data and a large amount of unlabeled data to reduce the annotation burden [14, 15].

Besides the issue of lack of annotations, another challenge in cell segmentation comes with the variety of image modalities. In addition to the most common H&E-stained images, phase contrast [16], differential interference contrast (DIC) [17], brightfield [18], and fluorescence [19] are the most frequently used image modalities in the diagnosis. Pathology images with different modalities play different roles in disease diagnosis and biology analysis. For example, non-invasive imaging methods such as phase contrast and DIC are needed in some diagnostic procedures to avoid the staining of cell specimens, which will lead to cell death [20, 21]. However, pathological images with different modalities have significant differences and inconsistencies in cell morphology, image illumination, contrast, color, cell size, etc. Because most existing cell segmentation methods only focus on a single modality, different segmentation methods need to be used for pathological images with different modalities, which is not only troublesome for deployment but also needs comprehensive parameter tuning and selection, making it time-consuming and complex. Therefore, if a unified cell segmentation framework can be designed for diverse modalities, the speed and efficiency of cell instance segmentation can be improved.

To solve the above problems, this paper proposes a framework for cell instance segmentation based on semi-supervised learning. The proposed framework can be simultaneously applied to multi-modality images, including brightfield, phase contrast, DIC, and fluorescence images, by only using a small number of fine-grained annotated images and a large number of unlabeled images. The contributions of this paper can be concluded as the following three aspects.

- A novel semi-supervised cell instance segmentation framework is designed for multi-modality pathological images. The modality of a given pathological image is judged by the unsupervised clustering method $K$-Means [22]. Images with different modalities are fed into a corresponding segmentation model for training and inference.

- The cell segmentation of multi-modality images is realized by an ensemble of Cellpose [23], U-Net [24], and CDNet [25]. Besides, two different semi-supervised learning strategies are adopted to fully utilize the unlabeled images to improve cell segmentation performance.

- Experiments show that the proposed framework can achieve an F1 performance above 0.80 on the tuning set, which ranks 6th among all teams in the challenge.

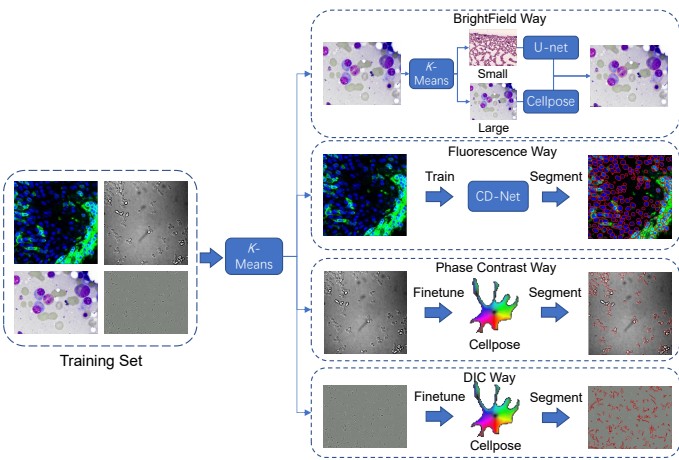

Figure 1: The overall architecture of our proposed framework for semi-supervised cell segmentation.

## 2 Method

### 2.1 Framework Overview

The challenge's objective is to solve the semi-supervised cell segmentation problem for four different image modalities, namely, brightfield, DIC, phase contrast, and fluorescence, under a unified segmentation framework. Considering the vast differences among the cell morphologies of the four modalities, it is difficult to solve the segmentation problem with a single model. To address this issue, we first utilize $K$-Means, an unsupervised clustering method, to classify the imaging modality. Then, different models are trained separately for each modality. The overall architecture of our framework is shown in Figure 1. There is one segmentation way for each modality, and in each way, we utilize one or two models for cell instance segmentation.

Considering that the cells in the brightfield images have varied cell diameters from approximately 20 pixels to 200 pixels, we further divide the brightfield images into two subcategories according to the cell diameter. A Cellpose model is utilized to segment the large-diameter subgroup, and a U-Net model is adopted for the small-diameter subgroup. For the fluorescence modality, the cells are small and dense, which makes cell segmentation quite similar to nucleus segmentation and hard for Cellpose to perform well. Therefore, we choose a state-of-the-art (SOTA) nuclear instance segmentation model called CDNet for fluorescence images. For the phase contrast and brightfield modality, which have medium cell shapes, we train two different Cellpose models. Different hyperparameters are tuned and utilized for each modality according to its particular cell morphology.

### 2.2 $K$-Means Clustering

Because manual annotations are strictly forbidden in the challenge, supervised models cannot be utilized to classify images with different modalities. Considering that the differences among the four modalities are enormous, it is possible to use unsupervised methods to realize classification. Specifically, $K$-Means is utilized for classification in our framework.

In the training phase, we transform the images in the training set (including labeled and unlabeled data because clustering does not require labels) from the RGB space to the HSV space. Then, the mean and variance of the HSV channels are calculated as the clustering features. Next, $K$-Means is adopted to find the center points of the features corresponding to these four modalities. In the testing phase, we calculate the feature vector of each input image and its Euclidean distance to the four centers. The cluster center closest to the feature vector of the input image is found, and the cluster is treated as the corresponding category of the input image.

## 2.3 Segmentation of Phase Contrast and DIC Images

In our framework, Cellpose is utilized to segment phase contrast and DIC images. Cellpose models are pre-trained on the Livecell dataset and can achieve SOTA segmentation performance on test images with very little user-provided training data, making Cellpose a general cell segmentation model that can provide excellent out-of-the-box results. More precisely, Cellpose composes a deep neural network with a U-Net style architecture and residual blocks and predicts a probability mask of a pixel inside a cell and the distances of pixels towards the center of a cell in $X$ and $Y$ axes. Eventually, the probability masks and the distances are fused to construct the cell instance masks.

Since Cellpose is pre-trained on the Livecell dataset, which is composed of phase contrast images, it shows extraordinary performance on the phase contrast images in the challenge. Besides, we find that DIC images can be segmented accurately after fine-tuning the Cellpose model. Thus, we directly use and fine-tune the Cellpose models for phase contrast and DIC Images.

However, as phase contrast and DIC images share similar color features, we find that $K$-Means has a decreased classification accuracy for them. To solve this problem, we additionally train a general model using the images from the phase contrast and DIC modalities. When the number of detected cells in the mask is small ($< 10$) after inference, we assume there is a high probability of misclassification, and the general model is used for inference again.

Besides, for Cellpose models, we use a pre-trained model to generate pseudo-masks for unlabeled images and then train models on unlabeled images with pseudo-masks. Finally, we fine-tune them with the labeled images as a cycle.

## 2.4 Segmentation of Brightfield Modality

For the brightfield modality, there are several challenges to realizing the segmentation. Firstly, the cell diameters in the brightfield images vary a lot. Statistically, the cell diameters of the brightfield modality distribute from approximately 20 pixels to 200 pixels. Secondly, cytoplasm and nucleus are stained separately with different colors, which significantly enhances the probability of misclassifying the nucleus as cells.

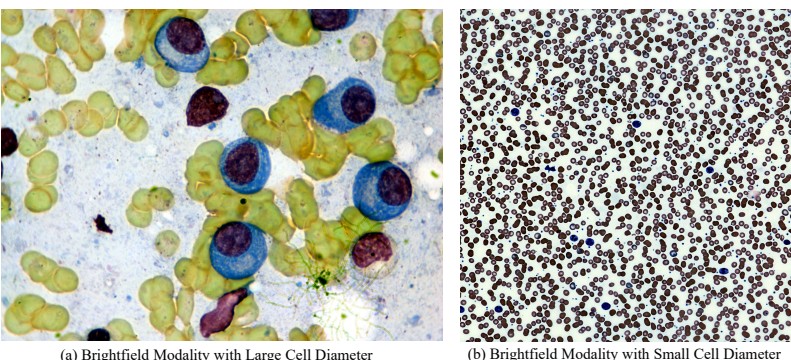

(a) Brightfield Modality with Large Cell Diameter     (b) Brightfield Modality with Small Cell Diameter

Figure 2: Varieties of different images of the brightfield modality. (a) is a brightfield image with a large cell diameter of approximate 200 pixels. The cytoplasm and nucleus are also stained with different colors. (b) represents a brightfield image with much smaller cells with around 20-pixel diameters. The blue-colored cell is surrounded by a large number of red, circle-shaped backgrounds.

In order to solve the problems mentioned above, we first divide the cell images into two subcategories with $K$-Means. The mean and variance of RGB values are treated as the features for clustering here. After clustering, we find that one cluster (subcategory) has a larger cell diameter and many cells overlap, as shown in Figure 2(a). Another subcategory is with smaller and more dispersed cells, as is shown in Figure 2(b). For the former subcategory, we choose the Cellpose model for segmentation, as we found that Cellpose performs better on large cells. For the other subcategory, U-Net is utilized as the segmentation model with VGG16 encoders. We utilize both the dice loss and the cross-entropy loss for training the U-Net. Technically, the loss of the U-Net is defined as:

$$L_{\text{U-Net}} = L_{\text{dice}} + L_{\text{ce}}, \tag{1}$$

where $L_{\text{dice}}$ and $L_{\text{ce}}$ represent dice loss and cross-entropy loss, respectively.

## 2.5 Segmentation of Fluorescence Modality

Fluorescence images are characterized by large cell number, small cell size, blurred cell borders, and high cell overlapping ratio, as shown in Figure 3. These features make cell segmentation in fluorescence images very similar to nuclear segmentation. CDNet [25] is a model for nuclear instance segmentation that constructs direction difference maps according to the centripetal direction feature of cell instances, aiming to learn the direction features of pixels pointing to the corresponding cell instance center. Therefore, CDNet can distinguish the overlapping cells and achieve SOTA performance in nuclear segmentation, and we adopt it for the cell segmentation of fluorescence modality.

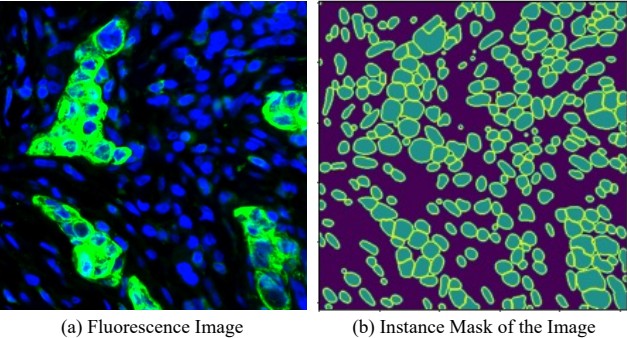

(a) Fluorescence Image        (b) Instance Mask of the Image

Figure 3: The image of fluorescence modality. (a) is the original fluorescence image. In the image, cell boundaries are blurred by black pixels. Besides, the green-colored cell cytoplasm overlaps the blue-colored cell nucleus. (b) is the corresponding cell segmentation instance map. The background is colored black, cell boundaries are colored yellow, and cell interiors are colored green.

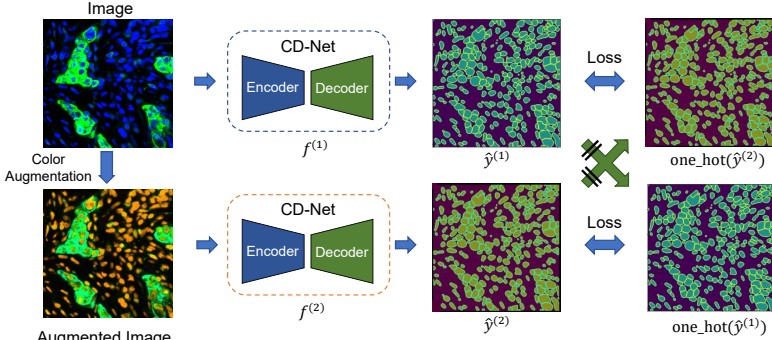

Figure 4: The cross supervision strategy for semi-supervised training.

To fully use the given unlabeled data of the fluorescence modality, we further develop a semi-supervised training strategy, as shown in Figure 4. Specifically, during the training phase, we initialize two CDNet instances $f^{(1)}$ and $f^{(2)}$ simultaneously. For the labeled images, we feed the training images into the two CDNets. Then, the dice and cross-entropy loss are utilized for full supervision between the prediction and the ground-truth mask as

$$L_l^{(i)} = L_{\text{dice}}(\hat{y}^{(i)}, y) + L_{\text{ce}}(\hat{y}^{(i)}, y), \; i = 1, 2, \tag{2}$$

where $\hat{y}^{(i)}$ stands for the predicted mask of the $i$th CDNet, and the ground-truth mask is denoted by $y$.

For unlabeled data, color augmentation is applied to the input images. Then, the pseudo-masks generated by the two CDNets are regularized by a cross-constraint regularity to train the other network.

We take the cross-entropy loss for the regularity, which is calculated by

$$L_u^{(1)} = L_{\text{ce}}(\hat{y}^{(1)}, \text{one\_hot}(\hat{y}^{(2)})), \tag{3}$$

and

$$L_u^{(2)} = L_{\text{ce}}(\hat{y}^{(2)}, \text{one\_hot}(\hat{y}^{(1)})), \tag{4}$$

where one_hot transforms the predicted probabilities into one-hot segmentation masks.

Finally, the total loss for fluorescence image segmentation is defined as

$$L_u^{(i)} = L_l^{(i)} + \alpha L_u^{(i)}, \, i = 1, 2, \tag{5}$$

where $\alpha$ is the weight of the loss for unlabeled data. It is set to $0.1$ in our experiment.

## 2.6 Inference Acceleration Strategy

Since the testing images are predicted by the docker one by one, the main time consumption during inference comes from importing packages and loading models, which takes about 9 seconds on average. To improve the inference speed, we do the clustering first, and then load the corresponding model, which means we only need to load one model instead of four and can accelerate the inference procedure.

# 3 Experiments

## 3.1 Dataset and Implementation Details

We use both the labeled and unlabeled data given by the challenge. To improve the generalizability of our models, we enlarge our dataset by adding part of the images from the Cellpose dataset [23] and the Ominipose dataset [26]. The development environments and requirements are presented in Table 1.

Table 1: Development environments and requirements.

| System | Ubuntu 18.04.5 LTS |
|---|---|
| CPU | Intel(R) Xeon(R) Silver 4216 CPU @ 2.10GHz |
| RAM | 16×4GB; 2.67MT/s |
| GPU (number and type) | One NVIDIA GeForce RTX 3090 |
| CUDA version | 11.4 |
| Programming language | Python 3.7 |
| Deep learning framework | Pytorch [27] (Torch 1.10, torchvision 0.11.1) |
| Code | https://github.com/shinning0821/nips_cellseg |

## 3.2 Environment Settings

### 3.2.1 Training Protocols

**Data Augmentation** When training the Cellpose models, no data argumentation is utilized. When training the U-Net and the CDNet, we use data argumentation, including affine transformation, random flipping, random blurring, and color jittering.

**Patch Sampling Strategy** In the training phase, we train the Cellpose models with the original images without considering the image size. For the U-Net trained on brightfield images, we crop all images to patches sized $512 \times 512$. For CDNet trained on the fluorescence images, the patch size is $256 \times 256$. In the inference phase, the patch size of the sliding window is the same as in the training phase for each model. Detailed training protocols of U-Net, CDNet, and Cellpose models are demonstrated in Table 2.

**Optimal Model Selection Criteria** For the Cellpose models, we use the default model selection criteria in Cellpose. For U-Net and CDNet, we split the training set, and 80% of the images in the training set are fed into the models for training. The remaining 20% images are used for validation. Then, the model with the best F1 score on the validation set is selected as the optimal model.

Table 2: Training protocols of U-Net, CDNet, and Cellpose. The number of parameters and FLOPs for Cellpose are unable to obtain due to the encapsulation of the Cellpose model.

| Method | U-Net | CDNet | Cellpose |
|---|---|---|---|
| Network initialization | random initialization | random initialization | pretrained on Livecell |
| Batch size | 8 | 16 | 8 |
| Patch size | $3 \times 512 \times 512$ | $3 \times 256 \times 256$ | the whole image |
| Total epochs | 20 | 20 | 100 |
| Optimizer | Adam [28] with nesterov momentum ($\mu = 0.99$) | Adam [28] with nesterov momentum ($\mu = 0.99$) | SGD |
| Initial learning rate (lr) | 0.0005 | 0.0005 | 0.01 |
| Lr decay schedule | halved by 20 epochs | halved by 20 epochs | halved by 20 epochs |
| Training time | about 2 hours | about 2.5 hours | about 10 minutes |
| Loss function | cross-entropy & dice loss | cross-entropy & dice loss | cross-entropy & dice loss |
| Number of model parameters | 20.27M | 20.47M | N/A |
| Number of FLOPs | 377440.17M | 148073.35M | N/A |

Table 3: Ablation studies on using unlabeled data for semi-supervised training.

| | Fluorescence Image | | Phase Contrast Image | | Brightfield Image | |
|---|---|---|---|---|---|---|
| add unlabeled data | ✗ | ✓ | ✗ | ✓ | ✗ | ✓ |
| mDice | 0.8773 | 0.8948 | N/A | N/A | N/A | N/A |
| mPrecision | 0.6860 | 0.7037 | 0.8728 | 0.8709 | 0.9296 | 0.9332 |
| mRecall | 0.7756 | 0.7767 | 0.8279 | 0.8390 | 0.8344 | 0.8366 |
| mF1-score | 0.7220 | 0.7320 | 0.8433 | 0.8484 | 0.8599 | 0.8621 |
| mInstDice | 0.7887 | 0.8084 | 0.8276 | 0.8344 | 0.8536 | 0.8560 |

## 4 Results and Discussions

### 4.1 Quantitative results on tuning set

The F1 score of our proposed framework on the tuning set is 0.8428, which ranks 6th among all teams. Ablation studies for the two semi-supervised strategies are conducted to show the effect of utilizing unlabeled data. The effects of utilizing unlabeled data for semi-supervised training are shown in Table 3. For the fluorescence modality, we generate patches from the unlabeled whole slide images. For the phase contrast and DIC images, we use the unlabeled patches given by the challenge. Furthermore, we find that the semi-supervised strategy works poorly on the brightfield images. Therefore we do not apply semi-supervised learning for brightfield, and only the labeled brightfield images are utilized. From the results, it can be concluded that unlabeled data slightly improves the performance for all three modalities. Specifically, the performance enhancement of the fluorescence images is much more evident than in other modalities.

### 4.2 Qualitative results on tuning set

Among the four modalities, the proposed model performs best on the fluorescence modality since the adopted CDNet and our semi-supervised training strategy deal with overlapped cells superiorly, as shown in Figure 5. Our model has a good performance for phase contrast and DIC modalities as well, as illustrated in Figure 6.

However, some things could still be improved for our framework. For example, the proposed framework performs poorly when dealing with cells that do not exist in the training dataset, as shown in Figure 7. Besides, the performance on the brightfield modality is not ideal as well. As demonstrated in Figure 8, some cells are ignored by our framework, and some nuclei are segmented as cells. This issue is due to the resolution difference between images in the training set and the

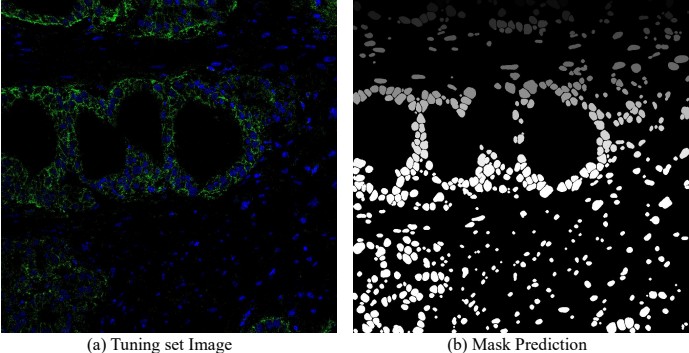

(a) Tuning set Image          (b) Mask Prediction

Figure 5: One example with good segmentation result. (a) is the original image that comes from the tuning set. (b) is the corresponding mask predicted by our model.

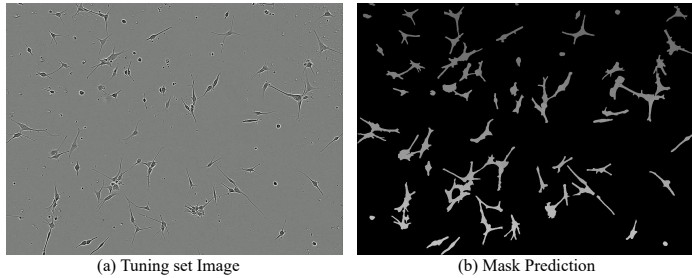

(a) Tuning set Image          (b) Mask Prediction

Figure 6: Another example with good segmentation result. (a) is the original image that comes from the tuning set. (b) is the corresponding mask predicted by our model.

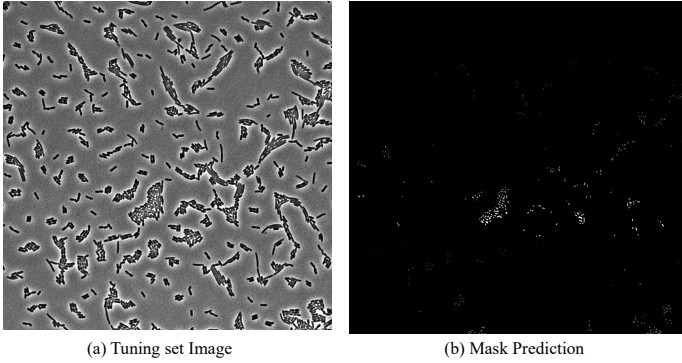

(a) Tuning set Image          (b) Mask Prediction

Figure 7: One example with poor segmentation result. (a) is the original image that comes from the tuning set. (b) is the corresponding mask predicted by our model.

tuning set, which leads to a significant difference in the cell diameter. There are some differences in cell morphology during training and validation as well, which poses significant challenges to the generalization of the model. Therefore, better solutions are still required to address these issues.

### 4.3 Segmentation efficiency results in the tuning set

For the small images, because we adopt the inference acceleration strategy, a more efficient inference can be obtained in practice. Overall, the average real running time for small images is around 20 seconds. For whole slide images, we use a sliding window inference strategy together with the

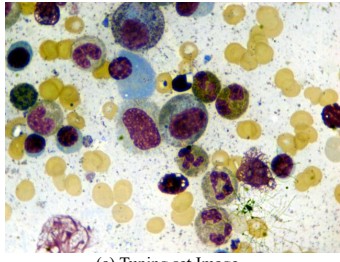 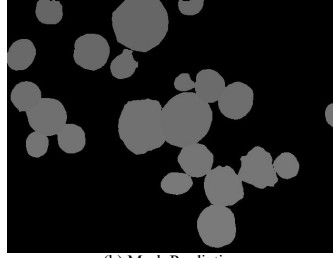

| (a) Tuning set Image | (b) Mask Prediction |

Figure 8: Another example with poor segmentation result. (a) is the original image that comes from the tuning set. (b) is the corresponding mask predicted by our framework.

Table 4: Results on final testing set

| | | Median F1 | | |
|---|---|---|---|---|
| Brightfield | DIC | Fluorescence | Phase Contrast | All |
| 0.9042 | 0.7036 | 0.0282 | 0.7626 | 0.7203 |
| | | Mean F1 | | |
| Brightfield | DIC | Fluorescence | Phase Contrast | All |
| 0.8983 | 0.6473 | 0.1058 | 0.6623 | 0.5686 |

inference acceleration strategy. By executing the running time evaluation script provided by the challenge organization on our machine (as shown in Table 1), we find that the inference time of the whole slide image in the tuning set is 162.05 seconds, which is within the time tolerance and proves that the proposed framework can segment the giga-pixel whole slide images efficiently.

### 4.4 Results on final testing set

The testing results of our model are shown in Table 4. From the result, we can conclude that our framework performs well on the DIC and phase contrast modality generally, which proves the effectiveness of the Cellpose model. However, it is unexpected that our model performs poorly on the fluorescence modality but excellently on the brightfield modality. In contrast, the segmentation result of the fluorescence modality is good and the results of brightfield images are not satisfactory on the tuning set. We think the poor performance on the fluorescence modality may result from the different domains of fluorescence images in the training, tuning, and testing set. However, the cell morphology in the brightfield modality is almost the same, and the proposed category subdividing strategy by $K$-Means according to the cell diameter advances the testing performance.

### 4.5 Limitation and future work

The testing results show that one of the limitations of the framework is that the segmentation performance drops sharply when the testing images are from a new domain which is not consistent with the training images. Therefore, in future work, domain adaptation methods are needed to be adopted to increase the generalization ability of CDNet.

## 5 Conclusion

This paper develops an ensemble framework to solve the cell instance segmentation problem for multi-modality microscopy images. Specifically, we design two semi-supervised training strategies to use the unlabeled data fully. Our framework shows good performance on brightfield images and most phase contrast and DIC images. In the future, more efforts should be paid to deal with the domain adaptation problem of fluorescence images, and the model generalization for phase contrast and DIC images needs to be improved as well.

## Acknowledgement

The authors of this paper declare that the segmentation method they implemented for participation in the NeurIPS 2022 Cell Segmentation challenge has not used any private datasets other than those provided by the organizers and the official external datasets and pretrained models. The proposed solution is fully automatic without any manual intervention.

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
