# OpenReview forum: "Semi-Supervised Cell Instance Segmentation for Multi-Modality Microscope Images"
_NeurIPS.cc/2022/Challenge/CellSeg — Submitted to NeurIPS CellSeg 2022_

### Official Review · Reviewer_XQbw · 2022-12-28
**Paper well structed, improvements needed**

**Rating:** 7
**Confidence:** 4

**Review:**

Summary:
The authors presented an ensemble framework to solve the cell instance segmentation problem for multi-modality microscropy images. K-Means was applied to select which segmentation network to use. By fine-tuning the Cellpose method, the cell segmentation of phase contrast, DIC, and bright field images is realized.

Pros:
The paper is well constructed.
Ablation studies confirm the effectiveness of the method with the semi-supervised learning strategy.

Cons:
The performance on Fluorescence images should be optimized and results should be further discussed.
Model generalization for phase contrast and brightfield images need to be further improved.

---

### Official Review · Reviewer_A9dU · 2023-01-13
**Well paper, but still need some modifications**

**Rating:** 6
**Confidence:** 4

**Review:**

## Summary
This paper proposed a semi-supervised cell instance segmentation framework for multi-modality microscopy. To address the issue that it is hard for single model to handle diverse modalities, the paper adopted a classification-segmentation strategy. Firstly, it classifies the input images by K-Means algorithm in HSV color space. After that, the corresponding model is selected for the inputs. The experiments verifies the effectiveness of the proposed framework on multi-modality cases.


## Pros:

+ The paper is well structed.

## Cons:

+ The authors may muddle the modalities. It is believed that the DIC Modality is actually Brightfield.

+ The writing should be strengthened. There are some typos in the paper, e.g. color argumentation in Section 2.4.

+ The paper did not provide efficiency results of WSI.

+ There is a general model for phase contrast and brightfield microscopy, therefore the framework may adopt two models during inference.

---

### Official Review · Program_Chairs · 2023-01-16
**Semi-Supervised Cell Instance Segmentation for Multi-Modality Microscope Images**

**Rating:** 9
**Confidence:** 4

**Review:**

Authors divided up the modality of the training images using k-means clustering prior to subjecting each to a specialist model. This is in contrast to the spirit of the competition.
To integrate unlabelled images during training, the authors came up with a cross constraint to regularize predictions using a paired augmented image. This allowed authors to successfully leverage the unlabelled data to improve models on average of ~0.01 across 3/4 of the modalities. The improvements were marginal but the method may still be of interest to the greater audience.
Implementation details and results were complete and final performance was impressive at 0.8428 on the tuning set.
Overall very well written.

---

### Official Review · Reviewer_QaYt · 2023-01-16
**Well paper**

**Rating:** 8
**Confidence:** 4

**Review:**

This paper presents a semi-supervised cell instance segmentation framework for multi-modality microscope images. Authors introduce effective methods for different sub-problems, e.g., K-Means for modality discrimination, Cellpose for cell segmentation, U-Net for sub-categories segementation, CDNet with semi-supervised learning for fluorescence segmentation. The overall method achieves very good performance, reaching f1 score of 0.8428.

---

### Author Response · Authors · 2023-02-14
**Point to Point Response for the Reviewers**


# Reviewer QaYt

**Response:** Thank you for your careful review and the positive comments about our paper.

# Reviewer NeurIPS 2022 Challenge CellSeg Program Chairs
**Comment 1:** Authors divided up the modality of the training images using $k$-means clustering. This is in contrast to the spirit of the competition.

**Response:** Thank you for your valuable comment! Due to the huge difference between different modalities, achieving high performance with a single model takes work. To achieve better performance, we cluster the input images into four modalities through $K$-Means and then get the segmentation masks through the corresponding segmentation model. Although this approach is a bit in contrast to the spirit of the competition, it obeys the competition rules as no manual annotations are introduced.

In addition, our study innovatively subdivides brightfield images into two subcategories according to cell diameter for better segmentation accuracy. Furthermore, our study adopts two types of semi-supervised learning, which takes full advantage of unlabeled data. Therefore, our study is significant for future research on semi-supervised multi-modality cell instance segmentation.

In the future, we will use a unified framework to achieve multi-modality cell instance segmentation so as to be more consistent with the spirit of the challenge.

# Reviewer A9dU
**Comment 1:** The authors may muddle the modalities. It is believed that the DIC Modality is actually Brightfield.

**Response:** Thank you for your valuable comment! Due to our negligence, we are sorry that the DIC modality is muddled with the brightfield modality in the paper. We have entirely proofread the manuscript and corrected the mistakes you pointed out.

**Comment 2:** The writing should be strengthened.

**Response:**
Thank the reviewer for reading the manuscript carefully! We have revised all possible grammatical errors and typos to our best in the manuscript. To list a few,

1) In Section 1, paragraph 3, line 2: the typo "**trainining**" is changed to "**training**";
2) In Section 2.1, paragraph 1, line 2: the typo "**contract** phase" is modified by "phase **contrast**";
3) In Section 2.4: as the reviewer has pointed out, the typo "color **augumentation**" is changed to "color **augmentation**";
4) In the title of Figure 3: the grammatical error "green-colored cell cytoplasm **are** extremely overlapped with the blue-colored cell nucleus." is changed to "the green-colored cell cytoplasm **overlaps** the blue-colored cell nucleus."

In addition to the grammatical errors and typos, we reorganize the paper's logical structure to improve the article's readability further.

**Comment 3:** The paper did not provide efficiency results of WSI.

**Response:** Thank you for your valuable comment! We use the running time evaluation code provided by the organizer to evaluate our docker's inference efficiency on the tuning set. Experiments show that on our machine (Intel Xeon Silver 4216 @2.10GHz with one Nvidia GeForce 3090 GPU), the inference time of the WSI in the tuning set is 162.05 seconds, which is less than the time tolerance. In the revised manuscript, we provide a detailed analysis of the efficiency results on WSI in Section 4.3.

**Comment 4:** There is a general model for phase contrast and brightfield microscopy, therefore the framework may adopt two models during inference.

**Response:** Thank you for your invaluable comment! Due to the similarity of these two modalities, it is difficult to distinguish them using $K$-Means accurately. So we use a general model to re-infer the segmentation masks of the images that may be misclassified, which will lead to poor cell instance segmentation performance. Although this strategy will lead to a more time-consuming inference process, the inference time consumption is still within the time tolerance. In the future, we will develop a more accurate unsupervised clustering method to decrease classification errors and avoid the use of general models, which will further improve the segmentation speed.

# Reviewer XQbw
**Comment 1:** The performance on Fluorescence images should be optimized and results should be further discussed. Model generalization for phase contrast and brightfield images need to be further improved.

**Response:** Thank you for your valuable comment! In the revised manuscript, we further analyze the performance of fluorescence images on the testing set. As shown in Section 4.4, the poor performance of our framework on fluorescence images may be caused by the inconsistency of the image domain between the training set, the tuning set, and the testing set. In the future, we will implement some state-of-the-art domain adaptation methods into our study to improve the generalization of the model on fluorescence, phase contrast, and brightfield modalities. The limitations of the current framework and future work are put forward in Section 4.5 of the revised manuscript.

---

### Decision · Program_Chairs · 2023-01-19

Accept